



# Impact of using a new ultraviolet ozone absorption cross-section dataset on OMI ozone profile retrievals

*Juseon Bak[1], Xiong Liu[1], Manfred Birk[2], Georg Wagner[2], Iouli E. Gordon[1], and Kelly Chance[1]*

*[1]Harvard-Smithsonian Center for Astrophysics, Cambridge, MA, USA*

*[2]Deutsches Zentrum für Luft- und Raumfahrt e.V. (DLR), Remote Sensing Technology Institute, Oberpfaffenhofen, D-82234 Wessling, Germany*

**Abstract**

We evaluate different sets of high-resolution ozone absorption cross-section data for use in atmospheric ozone profile measurements in the Hartley and Huggins bands with a particular focus on Brion-Daumont-Malicet et al. (1995) (BDM) currently used in our retrievals, and a new laboratory dataset by Birk and Wagner (BW) (2018). The BDM cross-section data have been recommended to use for retrieval of ozone profiles using spaceborne nadir viewing Backscattered UltraViolet (BUV) measurements since its improved performance was demonstrated against other cross-sections including Bass and Paur (1985) (BP) and those of Serdyuchenko et al (2014) and Gorshelev et al. (2014) (SER) by the "Absorption Cross-Sections of Ozone" (ACSO) activity. The BW laboratory data were recently measured within the framework of the ESA project SEOM-IAS (Scientific Exploitation of Operational Missions - Improved Atmospheric Spectroscopy Databases) to provide an advanced absorption cross-section database. The BW cross-sections are made from measurements at more temperatures and in a wider temperature range than BDM, especially for low temperatures. Compared to BW, BDM cross-sections are positively biased from ~2 % at shorter UV to ~5 % at longer UV at warm temperatures. Furthermore, these biases dynamically increase by up to ± 40 % at cold temperatures due to no BDM measurements below 218 K. We evaluate the impact of using different cross-sections on ozone profile retrievals from Ozone Monitoring Instrument (OMI) measurements. Correspondingly, this impact leads to significant differences in individual ozone retrievals by up to 50 % in the tropopause where the coldest atmospheric temperature is observed. Bottom atmospheric layers illustrate the significant change of the retrieved ozone values with biases of 20 % in low latitudes, which is not the case in high latitudes because the ozone retrievals are mainly controlled by a priori ozone information in high latitudes due to less photon penetration down to the lower troposphere. Validation with ozonesonde observations demonstrates that BW and BDM retrievals show altitude-dependent bias oscillations of similar magnitude relative to ozonesonde measurements, much smaller than those of both BP and SER retrievals.


However, compared to BDM, BW retrievals show significant reduction in standard deviation by up to 15 %,
especially at the coldest atmospheric temperature. Such improvement is achieved mainly by th better
characterization of the temperature dependence of ozone absorption.
1. **Introduction**
Accurate knowledge of the absorption cross-sections of ozone and their temperature dependence is
essential for highly accurate measurements of atmospheric ozone (Orphal et al., 2016) as well as other trace
gases affected by the strong ozone absorption such as BrO, $NO_2$, $SO_2$, and $CH_2O$ (e.g., Seo et al., 2019;
Theys et al., 2017). In the laboratory, measuring ozone cross-sections which can meet high requirements
for accurate ozone profile measurements is still challenging in covering a wide spectral range (at least 270-
340 nm) at high-resolution (at least 0.01 nm) at a wide range of atmospheric temperatures (180-300 K). The
difficulties range from reactivity of ozone to calibration standards. For instance, as discussed in the recent
review by Hodges et al. (2019) the accepted calibration of ozone cross-sections at mercury line (Hearn
1961) was in need of revision. In addition simultaneous measurements of ozone in microwave, infrared and
ultraviolet regions are subject to uncertainties in the respective regions (see discussion in Birk et al. (2019)
and Tyuterev et al. (2019) for instance). The need to evaluate existing cross-sections used for all
atmospheric measurements of ozone and to make its recommendations initiated the "Absorption Cross-
Section of Ozone (ACSO) activity" that was established in 2008 and conducted in two phases (2009-2011,
2013) (Orphal et al., 2016). The ACSO activity shows the need to continue laboratory ozone cross-section
measurements of highest quality.
Prior to ACSO activities, the available ultraviolet (UV) ozone-cross sections were thoroughly
reviewed by Orphal (2002, 2003) and as a result three datasets of ozone cross-sections were found to be in
agreement of 1-2 % with each other, including BP 1985 (Bass and Paur, 1985), BDM 1995 (Daumont et al.
1992; Brion et al., 1993; Malicet et al., 1995), and Global Ozone Monitoring Spectrometer (GOME) flight
model (Burrows et al., 1999) (GMFM). The BP dataset is no longer recommended for any atmospheric
ozone measurements (Orphal et al., 2016), but still used to keep the long-term consistency of ground-based
Dobson/Brewer total ozone records and spaceborne TOMS/OMI total ozone records (McPeters et al. 2015).
These cross-sections were also included in the 2004 edition of the HITRAN database (Rothman et al., 2005)
and remained unchanged in subsequent editions including HITRAN2016 (Gordon et al., 2017). Using
GMFM is restricted to GOME measurements because these cross-sections were measured at GOME
resolution (~0.2 nm). On the other hand, the high-resolution cross-sections of BDM were first applied by
Liu et al. (2005) for GOME ozone profile retrievals in the literature. In Liu et al. (2007), these three datasets
were thoroughly assessed to find the most suitable cross-sections for GOME ozone profile retrievals (290-



307 nm and 325-340 nm). As a result, they recommended using the BDM for ozone profile retrievals due
to much smaller fitting residuals and better agreement with ozonesonde measurements. Such improvement
is likely due to better spectral resolution and wavelength calibration of BDM than BP and GMFM. After
that, the recommendation of BDM for satellite ozone profile retrievals has been officially made by the
ACSO activities during the first phase (2009-2011) and the second phase (2013), respectively. The first
activity was focused on the intercomparison between BDM and BP, while the second activity was
additionally organized in response to the new publication of a high-resolution laboratory dataset covering
the temperature range of 193 to 293 K in 10 degree step by Serdyuchenko et al (2014) and Gorshelev et al.
(2014) (abbreviated as SER). In the framework of the ACSO activity, Liu et al. (2013) evaluated the impact
of changing from BDM to SER on Ozone Monitoring Instrument (OMI) ozone profile retrievals (270-330
nm). The recommendation of the BDM was made again for use in ozone profile retrievals. Recently, a new
laboratory dataset was measured at the German Aerospace Center (DLR) within the framework of the ESA
project SEOM-IAS (Scientific Exploitation of Operational Missions - Improved Atmospheric Spectroscopy
Databases) in order to improve the atmospheric BUV retrievals from the TROPOspheric Monitoring
Instrument (TROPOMI) on board the Sentinel 5-Precursor satellite (Birk and Wagner, 2018) (abbreviated
as BW). A publication with more details on experiment and analysis is in preparation. This motivates us to
investigate if the current recommendation could be replaced with the BW dataset. This work will also help
making decision on what cross-sections should replace BP measurements in the HITRAN database.
This paper is organized as follows: Section 2 compares the quadratic coefficients in the parameterization
of temperature dependence and evaluates the parameterized cross-sections against interpolated ones.
Section 3 analyzes the differences in individual OMI retrievals due to different cross-sections, which are
evaluated against ozonesonde observations in Section 4. This paper is finally summarized and discussed in
Section 5.

## 86   2. Comparison of BDM and BW

The BW dataset is publicly available at https://zenodo.org/record/1485588 along with some
experimental descriptions. A detailed publication is planned to describe the details of the experimental setup
and procedure so only a brief overview is given here. These cross-sections are given at six temperatures
(193, 203, 233, 253, 273, and 293 K) and at vacuum wavelengths in the spectral range 244 to 346 nm,
measured by means of Fourier-Transform Spectroscopy (FTS) at DLR at a spectral resolution of 3.3 cm$^{-1}$
(0.02-0.04 nm). A total of 191 measurements were recorded in two spectral ranges. Absorption cross-
sections were obtained at each temperature by means of a global least squares fit. Below 285.71 nm,
absorption cross-sections were smoothed to 7.7 cm$^{-1}$ (0.04-0.06 nm) resolution by convolving with a



Gaussian to reduce the noise. Offset corrections were made for each of the 6 temperatures by fitting to the
SER dataset since it was measured at higher ozone column density and thus considered more reliable
regarding offset. After offset correction polynomials of 1st order (<270.27 nm) > and 2nd order (>270.27 nm)
in temperature were fitted for each spectral point to improve the statistical uncertainty. The offset
corrections have minor effect on the cross-sections except for wavelengths above ~330 nm. Figure 1.a
illustrates BW measurements without polynomial fit in temperatures to be fairly compared with BDM
measurements (Fig. 1.b) with respect to the dependence of cross-sections on wavelength and temperature.
The BDM measurements are given at five temperatures (218, 228, 243, 273, and 295 K) and at air
wavelengths over the spectral range 195-519 nm with spectral resolution of 0.01-0.02 nm. Note that the
wavelengths of these measurements are converted to vacuum wavelengths in Figure 1.b.. Measured cross-
sections are typically parameterized quadratically to be applied conveniently at any atmospheric
temperatures using the following equation:
$$C = C_o + C_1(T - 273.15) + C_2(T - 273.15)^2 \quad (1)$$

The non-linear least squares fitting used in this paper converges typically within 3 iterations for both
BDM and BW. Measurements at 273 K are excluded for the BDM quadratic temperature fitting, according
to Liu et al. (2007). In Figure 2, the derived temperature dependent coefficients are illustrated, with their
relative differences. $C_o$ values are similar to each other in the Hartley band with relative biases of 2-3%.
However, the Huggins band shows large spiky biases of up to 8%. $C_1$ and $C_2$ represent linear and
quadratic temperature dependences of absorption cross-sections, respectively. The cross-sections in the
Hartley band are almost independent of the temperature variation and thereby large differences of these
coefficients between two datasets are due the large correlation between $C_1$ and $C_2$ and are of minor
importance to the parameterized cross-sections. However, the Huggins band shows the distinctly different
temperature dependence between two cross-section datasets, especially for the quadratic terms. For $C_2$, the
BW data show more monotonic wavelength dependence in the range 290-310 nm. Note that we determined
that the parameterization schemes used in this work and Birk and Wagner (2018) are very similar by the
fact that no residuals remain when comparing BW cross-sections with these two schemes (not shown here).
Figure 3 compares the residuals of the fitted cross-sections relative to the original measurements
interpolated to many atmospheric temperatures using a spline scheme. The BDM quadratic approximation
has large positive residuals of up to 15 % for the temperatures ranging from 243 and 295 K due to
insufficient sampling to account for the non-linearity of the temperature dependence, especially for the
longer UV wavelength range. Moreover, approximating the BDM cross-sections at temperatures below 218
K results in errors of ± 5% below 315 nm and up to ± 40% above. Compared to the BDM dataset, the


parameterization of BW cross-sections results into significantly reduced residuals, of 0.25% below 320 nm
and typically less than 2% at longer wavelengths if the temperature is within the boundaries of the
measurements. Residuals are within 5% even if the temperatures are out of the boundaries. This
demonstrates that the temperatures of BW measurements are well selected to characterize the temperature
dependence of ozone cross-sections, whereas cross-section errors due to the BDM parameterization exist.
Figure 4 shows the direct comparison of parameterized cross-sections between BDM and BW. The
difference of cross-sections between BDM and BW are generally consistent with the corresponding
comparison of $C_0$ around 270 K. The differences at different temperatures are typically within 2% for
wavelengths below 310 nm except for several spikes around 276, 297, and 306 nm that are correlated with
the differences of $C_2$. At wavelengths larger than 315 nm, the biases show large temperature dependence,
with the bias range increasing from ~5% at 315 nm to ~20% at 340 nm.

### 3. Impact of using different cross sections on ozone profile retrievals

OMI ozone profiles are retrieved at 24 layers from BUV spectra for 270-309 nm in UV1 and 312-330
nm in UV2 using an optimal estimation technique (Liu et al. 2010). The implemented configurations in this
work are similar to those in Liu et al. (2013). One orbit of measurements on 1[th] July 2006 is used to see
how our retrievals are changed due to using different cross-sections. Figure 5 shows the response of our
retrievals to the parameterization errors shown in Figure 3 as functions of solar zenith angle (SZA).
Compared to the BDM, the ozone retrievals are almost independent of the BW parameterization errors,
with individual differences of 2-3% below 20 km and ~0% above. The differences of the BDM cross-
sections with and without the parameterization are -5 to 15% in the lower troposphere at smaller SZAs and
up to ± 20% around 10 km at higher SZAs. The UV photon penetration down to the lower atmosphere
decreases with SZAs increasing and thereby tropospheric ozone retrievals become insensitive due to cross-
section errors at high SZAs, while a priori ozone information becomes more important to the retrieval.
Figures 6-8 show the retrieval differences when parameterized BW and BDM cross-sections are
implemented, respectively. To evaluate the different implementations, both fitting and retrieval accuracies
are assessed. However, it is very hard to see large differences in fitting residuals at final iteration compared
to differences on the retrieved elements of the state vector because the algorithm iteratively updates the
state vector toward minimizing the differences in the spectral residuals. The fitting residuals are comparable
at final iteration when applying BW and BDM dataset as shown in Figure 6.a except for noticeable smaller
residuals in 310-320 nm. However, we can find the distinct change in the mean residuals of measured
radiance to simulated radiance at initial iteration, mainly over the wavelength range of 290 to 315 nm, up
to 5 % as shown in Figure 6.b. On the other hand, Liu et al. (2007, 2013) demonstrated the distinct change





of final fitting residuals when changing BDM to BP and GMFM, implying that using BW dataset improves
fitting accuracies over using BP and GMFM, but produces similar fitting accuracies to using BDM and
SER. Figure 7 shows relative differences of the retrieved ozone profiles with the corresponding temperature
profiles taken from the National Centers for Environmental Protection (NCEP) final (FNL) operational
global analysis data. Large differences of 20-50% commonly exist along the tropopause, where the original
BDM measurements could not cover atmospheric temperatures below 218 K. Some larger differences occur
throughout the troposphere in the tropics likely due to the relative smaller retrieved partial ozone columns.
The individual differences of retrieved ozone in the lower troposphere are ~ 20%. Corresponding
differences of total column ozone, from integrating retrieved ozone profiles, are also presented by the black
line in Fig. 7a. Applying BDM causes an underestimation of total ozone except at the South Pole, despite
the overestimation being prominent for the individual layer columns in the troposphere. The magnitude of
this underestimation/overestimation is ~1 %, which is comparable to the overall accuracy (~1.5%) of the
OMI operational total ozone product against ground-based measurements (McPeters et al. 2015). The
wavelength shifts between ozone cross-sections and radiances are iteratively and simultaneously fitted with
ozone for their respective UV1 and UV2 channels. Figure 8 compares how the wavelengths of different
cross-sections are adjusted in each fitting window at nadir view. According to Schenkeveld et al. (2017),
wavelength errors of OMI radiances are expected to be ~0.002 nm in UV2 and ~0.015 in UV1. The fitted
wavelength shifts fall in the ranges of the OMI wavelength accuracy. Compared to the BDM, the BW
dataset has the relative shifts of ~0.002 nm in the UV2. The mean shifts in the UV1 are comparable, 0.0087
nm and 0.0081 nm for BDM and BW, respectively, whereas the variance of the fitted shifts over the latitude
is reduced with the use of BW dataset as the shifts are more stable south of 30°S. On the other hand, Liu et
al. (2013) shows that the relative shifts between SER and BDM are ~ 0.007 nm in both UV1 and UV2, and
BP shifts vary largely with latitude by up 0.01 nm. These results indirectly demonstrate the similarity of the
wavelength calibration quality between BDM and BW measurements.

**4. Validation with ozonesonde observations**
Ozonesonde measurements at five stations during the period 2005 to 2008 are used to evaluate the
retrieval accuracy of ozone profile retrievals using different cross-sections. In addition to the currently used
BDM and the new BW datasets, BP and SER previously assessed in Liu et al. (2013) are included in this
evaluation. Typically, high-resolution vertical structures of ozonesonde profiles (~100 m) are degraded to
OMI resolution (6-10 km in the stratosphere, 10-15 km in the troposphere) using retrieval averaging kernels
to eliminate the effect of OMI smoothing errors (80% of total retrieval errors in the lower stratosphere and
troposphere) in comparison with ozonesondes; as a result, the standard deviations of comparisons are

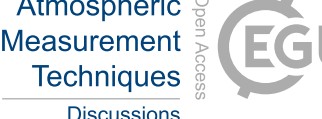

typically reduced by a factor of 2 in the troposphere and lower stratosphere while the comparisons of mean
biases are less impacted by using OMI smoothing errors or not. In this paper, the conclusion on which cross-
section data should be used stays the same no matter whether ozonesonde profiles are vertically smoothed
or not, so we present validation results only using original ozonesonde measurements. In Figure 9, mean
biases of the retrieved ozone profiles relative to ozonesondes and the corresponding standard deviations are
presented at each station, arranged by latitude from north to south, together with corresponding temperature
profiles.

In layers above ~20 km, a negligible impact of using different cross-sections is found because the

measurement information comes mainly from the Hartley ozone absorption band with little dependence on
temperature variation. Both BP and SER measurements provide a wider temperature range and more
samplings than BDM, but switching from BDM to BP / SER results in large altitude-dependent oscillations
of mean biases below ~20 km and noticeably fewer successful retrievals, consistent with Liu et al. (2013).
These oscillations tend to be wider with the minimum atmospheric temperatures decreasing such that the
mean biases increase ±50% at mid/high latitudes (210-215 K) to ±70% at low latitudes (200-205 K), which
is partly due to smaller ozone concentration in the tropics and hence the larger relative differences. This
result implies a defect in accounting for the temperature dependence in both the BP/SER cross-section
datasets, especially in the lower temperature range. Using BDM and BW cross-sections generally show
much smaller altitude-dependent oscillations of mean biases. The magnitudes of the biases are smaller for
BDM for the two middle/high latitude stations, but smaller for BW at the other, lower latitude stations. The
BW retrievals typically show negative biases of up to 30% relative to BDM retrievals. The number of
successful BW retrievals is slightly smaller than that of BDM retrievals because the negative biases cause
more occurrences of negative ozone so that the retrieval convergence is more difficult. It is difficult to
determine which one is better for ozone profile retrievals from the mean biases as OMI radiances contain
systematic radiometric calibration errors (Liu et al., 2010) and ozonesonde observations can also contain
systematic measurement errors (Liu et al., 2006).

As seen from the comparison of standard deviations in the middle panels, the use of BW consistently

gives significantly smaller standard deviations, by 5-20% in the lower stratosphere and upper troposphere
except for the high latitude station, Sodankyla. BW, BDM, and SER retrievals show similar standard
deviations at this station probably due to relatively warmer temperature, ~210-220 K in this altitude range.
In Figure 10, individual differences of layer column ozone between OMI retrievals and ozonesondes using
BDM and BW datasets are plotted as a function of temperatures for 8 layers below ~20 km. In this
comparison, the noticeable reduction of the scatter between OMI and ozonesonde, by 5-15% at layers from
17 to 8.5 km as well as by a few % below or above them, after applying BW cross-sections is further evident.
Improvements of the retrieval precision corresponding to standard deviations have been less often achieved
than those of the retrieval accuracy corresponding to mean biases; for examples, systematic errors in ozone
profile retrievals could be reduced by accounting for polar mesospheric clouds (Bak et al. 2016) and slit
function errors (Bak et al. 2019) as well as applying empirical calibration (Bak et al. 2017) whereas the
reduction of the standard deviations was achieved only in Bak et al. (2013) by better representing
dynamically induced ozone variability in the a priori ozone. This significant improvement in standard
deviations indicates that temperature dependence is better characterized at the lower temperatures near
~200K by the BW dataset.

## 5. Summary and discussion

This paper evaluates the recently measured laboratory high-resolution BW (2018) ozone cross-section data
within the framework of the ESA project SEOM-IAS to see whether or not the current recommendation
could be changed for improving ozone profile retrievals from UV measurements. The BDM (1993) dataset
has been regarded as the standard ozone absorption cross-section in space-based ozone profile retrievals
from BUV measurements: thereby we focused on comparing BW and BDM datasets and their impact on
our ozone profile retrievals from OMI BUV measurements. Compared to BDM given at 5 temperatures
ranging from 218 to 295 K, the BW dataset provides improved temperature coverage of 193 to 293 K, every
20 K. To conveniently apply the cross-section measurements at any temperature, we quadratically
parameterized its temperature dependence using iterative non-linear least squares fitting. The 273 K
measurements are excluded in the BDM parameterization to improve the fitting residuals at other
temperatures. However, the BDM parameterization causes increasing biases in approximate cross-sections
at lower temperatures using their 243 and 218 K measurements, especially at longer wavelengths in the
Huggins band (up to 20%). It reveals serious errors of up to ± 40% in representing the values at lower
temperatures out of the BDM measurements. In comparison, the BW approximation is very closely
parameterized to the original data, typically within 2%, while most of the atmospheric temperatures are
covered by the BW dataset; the biases increase to ±5% at temperatures below 195 K. Correspondingly,
individual ozone profile retrievals show less sensitivity due to the BW parameterization errors, with biases
of ~ 2% or less over the altitude range. On the other hand, using the parameterized BDM causes biases of
5-10% at bottom layers in the low latitudes and 10-20% at the tropopause. Relative to the BDM dataset, the
BW data show systematic biases of 2-3% in $C_o$ at shorter wavelengths below 300 nm, but larger spikey



biases of up to 8% at wavelengths longer than 315 nm. The difference in $C_1$ and $C_2$ implies distinctly
different temperature dependence especially in non-linearity in the Huggins bands. We then compared
ozone profile retrievals from one orbit of OMI measurements with BW and BDM cross-section datasets.
Using different datasets gives comparable results in the wavelength shift of cross-sections relative to OMI
radiance wavelengths and fitting residuals at the final iteration, respectively. However, the initial iteration
gives ~5% differences in fitting residuals near 290-315 nm, which results in significant differences of the
adjusted ozone profiles at the final iteration, ~50% at the tropopause across most latitudes and ~20% at the
bottom layers in the low-latitudes. To evaluate the quality of ozone retrievals, ozonesonde measurements
are compared at five stations. In this validation, we include other cross-section datasets, BP (1985) and SER
(2014). Compared to the large vertical oscillation of mean biases for OMI ozone profiles using BP and SER,
the BW retrievals show mean biases comparable to or sometimes improved over the BDM retrievals. The
most important improvement due to switching from BDM to BW is the significant reduction of the standard
deviations, by up to 15% in the lower stratosphere and upper troposphere where atmospheric temperatures
are lower than ~200K.

Based on this evaluation, switching our ozone absorption cross-section reference from BDM to BW is

very promising for OMI ozone profile retrievals. However, in this evaluation soft calibration is turned off
and thereby the final decision on our algorithm will be made after further evaluating our retrievals with
BW-based soft calibration. In order to make a robust recommendation it might be useful for the ACSO
committee to organize another activity to assess the impact of applying this new dataset on other ozone
measurements on column ozone or profiles from various platforms. The results of this work in addition to
that of Orphal et al. (2016) will help the HITRAN committee to decide which cross-sections should be
included in HITRAN2020 edition.

Using different ozone cross-sections could also cause an important change in $SO_2$ retrievals fitted in the

Huggins band and therefore it the impact of applying both ozone and $SO_2$ cross-sections available from the
BW datasets (https://zenodo.org/record/1492582) should be evaluated. However, the spectral coverage of
the BW dataset is insufficient for the spectral fitting of other trace gases such as BrO and HCHO, both of
which have significant interference with ozone. Ozone cross-sections in other wavelength ranges, such as
the mid-infrared region near 9.6 μm and the Chappuis band (400-650 nm), have not been thoroughly
evaluated in the literature. The ozone profile algorithm used in this work will be implemented for the
Tropospheric Emissions: Monitoring of Pollution (TEMPO) satellite combining the UV and visible
measurements to improve the detection of boundary layer ozone. Therefore we should extend this work to
find the most suitable ozone cross-sections in the TEMPO visible ozone channel (540-740 nm), focusing



on SER 2014 covering from 213 to 1100 nm (193-293 K in 10K steps) and that of Brion et al. (1998) which
provides measurements at 218 and 295 K from ~520 nm to ~650 nm. Moreover, the need to improve wide
spectral range laboratory cross-section measurements of ozone is still required to advance atmospheric
ozone and other trace gases measurements.
*Author contributions*. JB and XL designed the research; MB and GW provided oversight and guidance for analyzing
cross-section dataset; XL contributed to analyzing ozone profile retrievals; JB conducted the research and wrote the
paper; IG and KC contributed to the analysis and writing.

*Competing interests*. The authors declare that they have no conflicts of interest.

*Data availability*, The BW cross-section dataset is available at https://zenodo.org/record/1485588. OMI
Level1b radiance datasets are available at https://aura.gesdisc.eosdis.nasa.gov/data/Aura_OMI_Level1/
(last access: 31 Nov 2019). The ozonesonde data used to validate our ozone profile retrievals were obtained
though    the    WOUDC    and    SHADOZ.    The    WOUDC    dataset    is    available    at
https://woudc.org/data/products/ozonesonde/(last access: 31 Nov 2019) and for the SHADOZ dataset at
https://tropo.gsfc.nasa.gov/shadoz/Archive.html (last access: 31 Nov 2019).

*Acknowledgement*, We acknowledge the OMI science team for providing their satellite data and the
WOUDC and SHADOZ networks for their ozonesonde datasets. Research at the Smithsonian Astrophysical
Observatory by Juseon Bak, Xiong Liu, and Kelly Chance was funded by the NASA Aura science team
program (NNX17AI82G). MB and GW thank the European Space Agency (ESA) for funding of the SEOM-
IAS project (ESA/AO/1-7566/13/I-BG).

*Financial support*. This research has been supported by NASA Aura science team program (grant no.
NNX17AI82G). The SEOM-IAS project has been funded by ESA (ESA/AO/1-7566/13/I-BG).

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

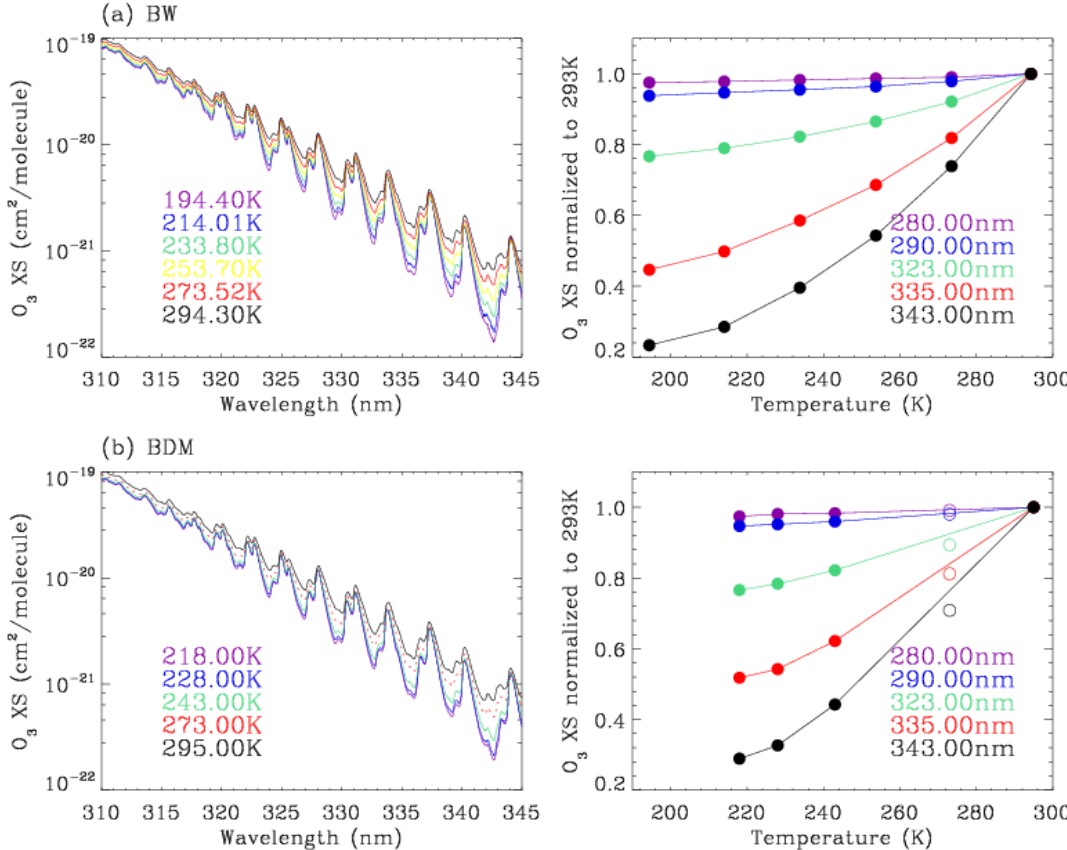

**Figure 1. (Left) Measurements of ozone absorption cross-sections at all selected temperatures in the Huggins bands taken from (a) BW (2018) and (b) BDM (1995), respectively. (Right) For BW, the experimental data are plotted without the quadratic parameterization for a fair comparison with BDM. BDM measurements at 273 K are plotted with a dotted line on the left and with open circles on the right, because the data at this temperature are not recommended for use, by Liu et al. (2007).**



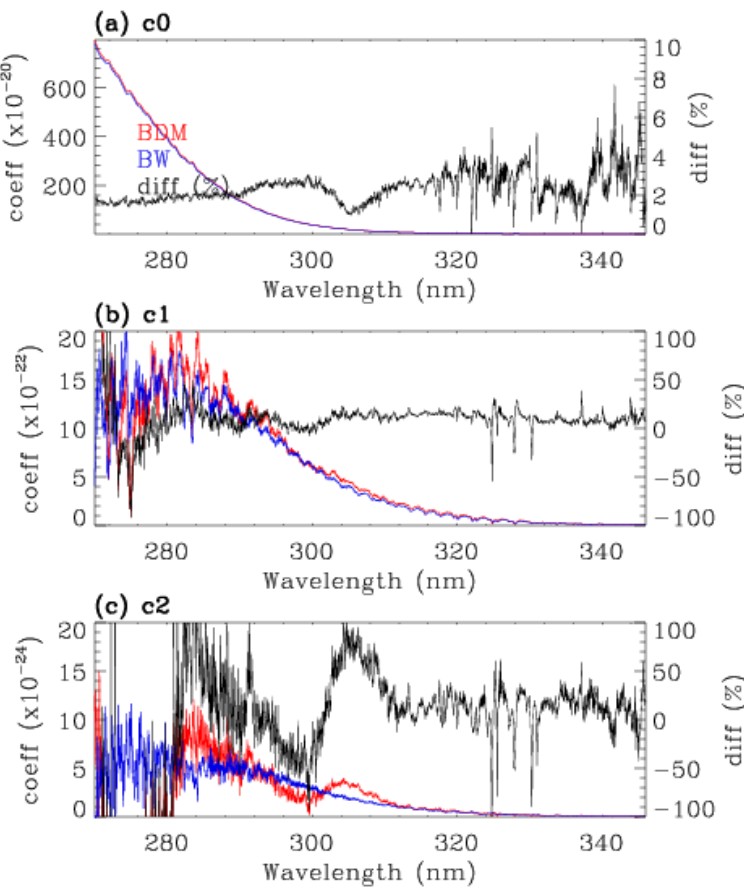

421

**Figure 2. Quadratic coefficients (cm²/molecule) to parameterize the temperature dependence of ozone cross-sections for BDM (red) and BW (blue), respectively, with their relative differences (BDM-BW)/BW in black.**



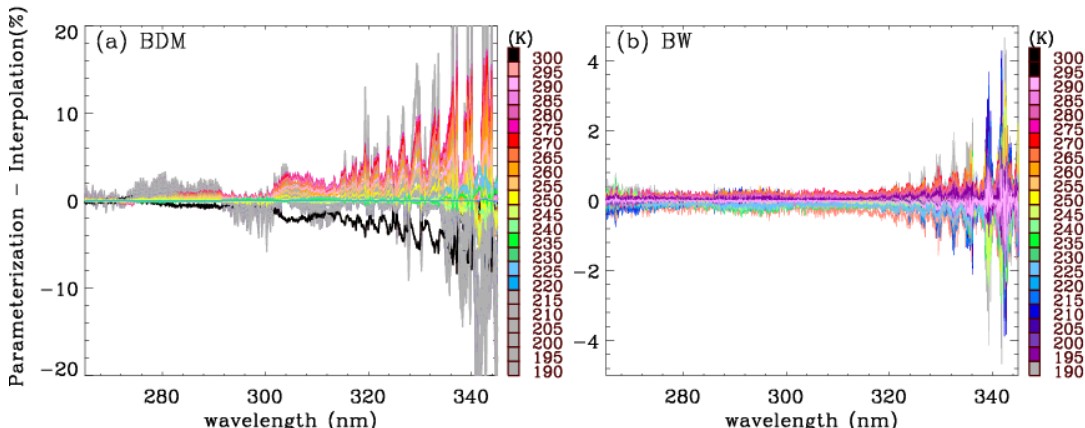

**Figure 3. Relative differences of ozone cross-sections parameterized and spline interpolated at temperatures between 190 and 300 K, for (a) BDM and (b) BW, respectively. In the legend, the temperatures not covered by each dataset are indicated with gray and black, for values beyond lower and upper boundaries, respectively.**

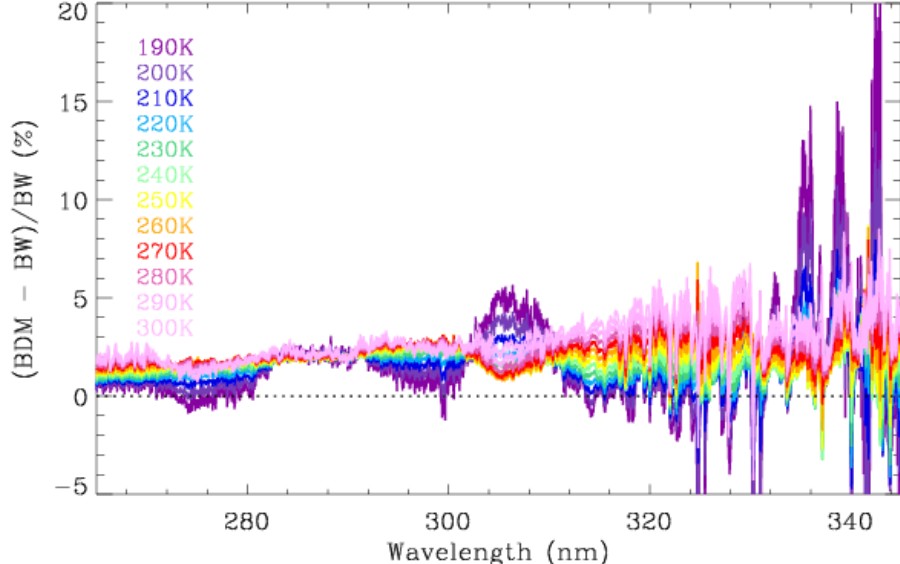

**Figure 4. Same as Figure 3, but for relative differences (%) of parameterized ozone cross-sections between BDM and BW.**



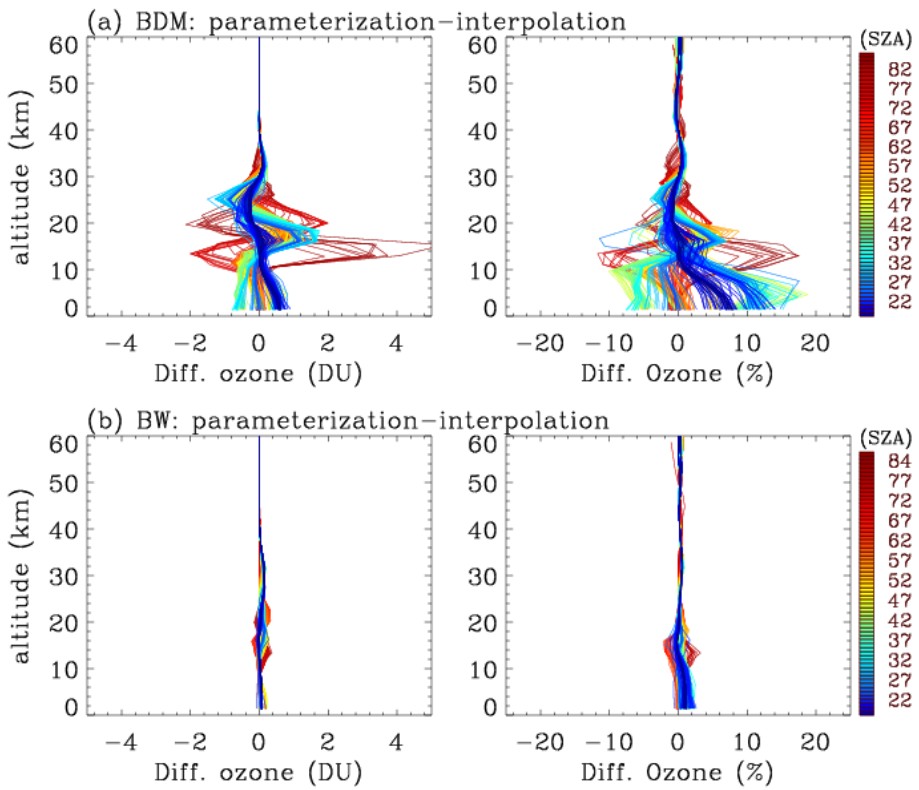


**Figure 5. The impact of parameterizing the cross-sections shown in Figure 3 on ozone profile retrievals, for (a) BDM and (b) BW.**







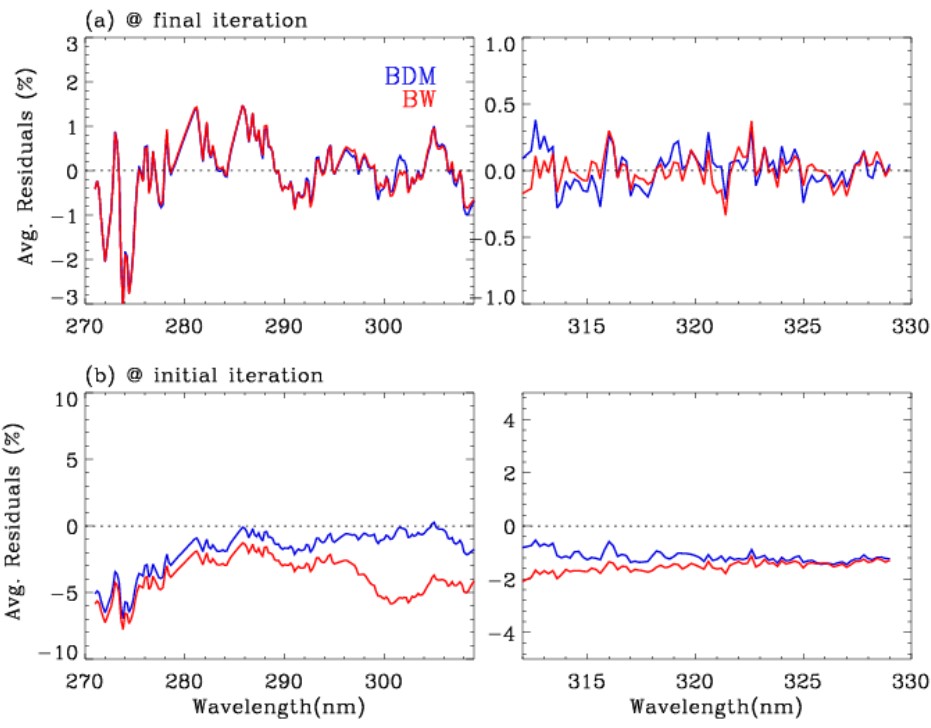


**Figure 6. Comparison of mean fitting residuals at latitudes of 15° S to 15° N at (a) final iteration and (b) initial iteration, respectively, when using BDM (blue) and BW (red).**





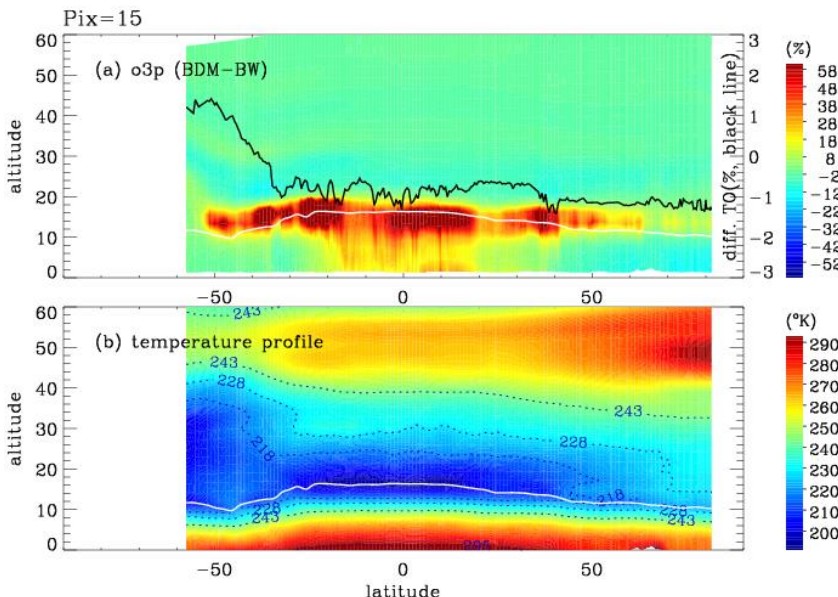


**Figure 7. (a) Percent Difference ((BDM-BW)/BW x 100%) of retrieved ozone profiles using BDM and BW datasets at nadir view, and (b) corresponding temperature profiles in the retrievals. In the upper panel, the black line represents the differences of integrated column ozone. The white line in both panels represents the tropopause height.**



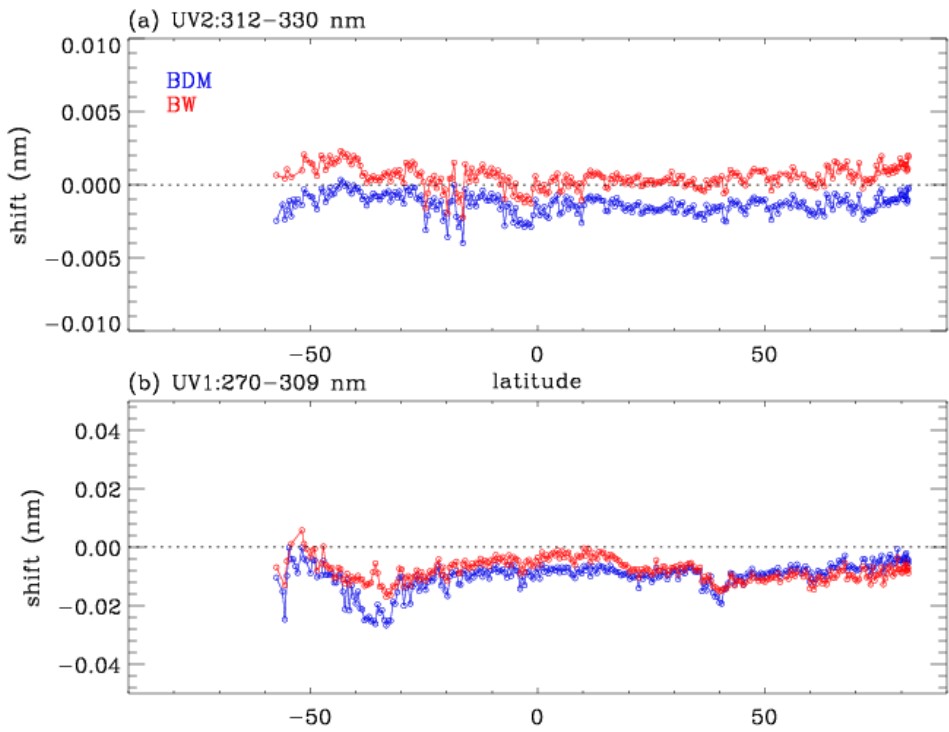

**Figure 8. Comparison of the wavelength shifts (nm) between ozone cross-sections and OMI radiances at the nadir view for using BDM (blue) and BW cross-sections, respectively.**

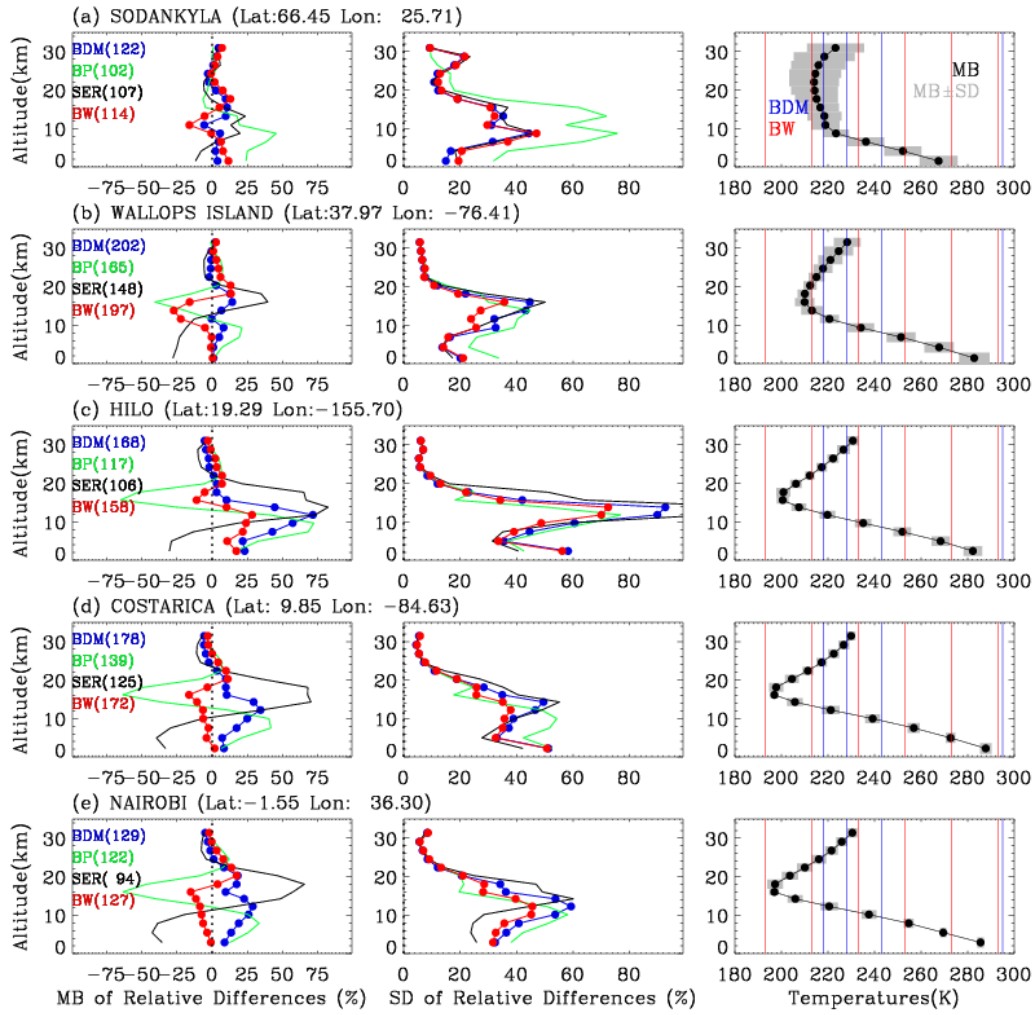

452

**Figure 9. (Left column) Mean biases of relative differences between OMI and ozonesonde ozone profiles at five stations arranged with decreasing latitude when four different cross-sections are applied to OMI retrievals, with (Middle column) the corresponding standard deviations and (Right column) mean temperatures (black circle) of individual profiles (gray). The numbers after the four cross-sections in the legends show the number of successful retrievals. Blue and red vertical colors in the right panels represent the temperatures used to derive the quadratic coefficients from BDM and BW measurements, respectively.**

460

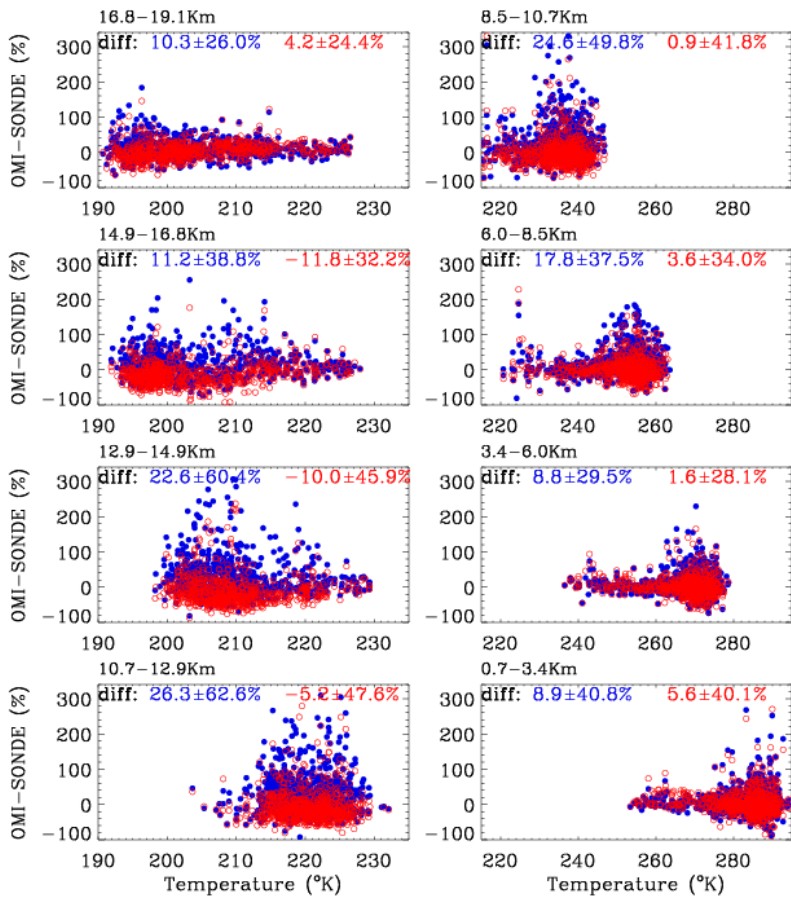

461

**Figure 10. Scatter plots of individual differences between OMI retrievals using BDM (blue) and BW**

**(red) cross-sections and ozonesonde measurements for each layer from the surface (bottom right)**

**to 19.1 km (top left) as functions of layer temperature. Mean differences and standard deviations**

**for both cross-sections are shown in the legends.**

466