# Peer review of "Impact of using a new ultraviolet ozone absorption cross-section dataset on OMI ozone profile retrievals"

_Atmospheric Measurement Techniques, 2020_

## Referee Comment (RC1) · Anonymous Referee #1 · 26 Apr 2020

**General comments:**

This manuscript, AMT-2020-94, reports the evaluation of new O3 xsec data sets (labeled as 'BW') measured in the Hartly and Huggins bands for the use of O3 profile retrieval from OMI observations. The BW data sets were modeled by using a polynomial in a function of temperature in order to facilitate direct comparison with the current reference data set ('BDM') and their application to the O3 profile retrieval. They have found that the new data set, BW, shows a better performance in the retrieval of O3 profile in terms of less oscillatory features in the retrieved profile and better agreement with the ozonesonde data. We found the manuscript written in a nice and compact manner; the presentation looks consistent. However, we're not convinced that we can agree with the authors' interpretation of what is described in Sec. 2, which will be detailed below.

This manuscript has shown well that the new dataset, BW, is better than the BDM in the O3 profile retrievals primarily because of their wider temperature coverage, esp. going down to 194 K critical to the retrievals in the transition layers (UTLS), which was not covered by the BDM data set in temperature. Therefore, the conclusion of this work has bee supported by the results presented in the manuscript. The topics of this paper highly relevant to the scope of AMT, so that we recommend a publication of this manuscript to AMT with a revision or a further clarification Sec. 2. Specific comments and suggestions follow.

The authors wrote, "*Offset corrections were made for each of the 6 temperatures by fitting to the SER dataset since it was measured at higher ozone column density and thus considered more reliable regarding offset.*"

→ Does this mean that the BW xsec was 'normalized' to that of SER? Clarify what the corrections factors were and how (and what wavelengths) they were determined. Was this offset considered in the error budgets?

Authors wrote, "*After offset correction polynomials of 1st order (<270.27 nm) > and 2nd order (>270.27 nm) in temperature were fitted for each spectral point to improve the statistical uncertainty.*" and followed by "*Measured cross-sections are typically parameterized quadratically to be applied conveniently at any atmospheric temperatures using the following equation: $C = C_0 + C_1(T - 273.15) + C_2(T - 273.15)^2$          (Eq.1)*"

→ (1) The agreement between the original data and the fitted data should be inspected or discussed for each of the two data sets, BW and BDM, and discussed. Besides, direct comparison of their original data sets between BW and BDM (prior to having them fitted to the polynomial), which may be done at T = 273 and 295 K provided that their temperature differences, $\Delta T = 0.5$ and 0.7K, respectively, is insignificant, which seems true because the authors argued the dominant coefficient C0 is almost independent of temperature.

→ (2) We're not sure how well the Eq. (1) could have captured the temperature dependence of the xsec. The xsec can be represented by integrated (line) intensities for the given frequency (wavelength) grid, and the temperature dependence of the line intensities can be modeled by two parameters, i.e., partition function (which we know well for O3) and the lower state energies (which we do not know for the features of this work). Thus, one can simulate the intensity ratio

to that at 296 K at various temperature for a few representative cases of the lower state energies, as shown in Fig. X below.

[Figure]

**Fig. X** S/S(296K) vs. T at a given E"(cm$^{-1}$) =[100,400,700,1000, 1300, etc.]

As we see, Fig. X is similar to the right panel of Fig. 1, except for one thing that each curve in Fig. X represents different values of the lower state energies, not the wavelength presented in Fig. 1. There is a possibility of having the sampled wavelengths (such as 280, 290,..., in nm) possessing progressively higher value of their (effective) lower state energies. However, it'd be more appropriate to assume that each curve in Fig. 1 corresponds to a different value of 'effective value of the lower state energy' of multiple transitions falling into the particular wavelength data point grid (for instance, 280nm±resolution element). This point should be addressed properly to keep naive readers from being misled to think the temperature dependences in Fig. 1 is attributed to the wavelengths.

→ (3) For the same reason, Fig. 2 is hard to interpret. The respective comparison of the C1 and C2 for two different data set as a function of nm could be legitimate only when the two data sets are measured at the same resolution because the effective lower state energies mentioned above would be the same. Therefore, the non-wavy feature of C2 for the BW data set would have more to do with the outcome of the resolution choice in the representation by Eq.(1), rather than it is telling the BW data set is superior to the BDM dataset in the temperature consistency. In other words, Fig. 2 shows which data set is better represented by Eq. 1 rather than which data is closer to the truth.

This section may stay, but with a specific statement being provided for the readers on the point made above. The bottom line is that the BW data set is better than the BDM set because of the broader coverage of the measurement temperature, especially covering the temperature critical to UTLS layers, as was properly concluded by the authors in the manuscript.

End of document.

---

## Referee Comment (RC2) · Anonymous Referee #2 · 18 May 2020

**1    General**

The manuscript AMT-2020-94 provides a comparison of UV ozone retrievals from the OMI instrument using a new cross section data set (BW, provided in the frame of the ESA SEOM-IAS project) with the standard data set from Reims (BDM). Overall, the manucript is very well written, nicely structured and argued. Selected figures do well illustrate the discussion in the manuscript. The presentation is scientifically sound and clear. The topic fits nicely within the journal scope and, therefore, I can fully recommend publishing the manuscript.

There are a few issues to the current paper that need to be addressed before publica-

tion, however.

1. The analysis is based on a new cross section data set (BW data) that at this point of time is openly available, but has not yet been published in the scientific literature. It therefore lacks yet the scrutiny of the peer-review process. While this is a regrettable fact, it does not invalidate the present work. But the authors must carefully discuss what might possibly be an inherent contradiction. In a previous study (Liu et al., 2013), the authors have concluded that another recent UV cross-section data set (the SER data from Bremen, Serdyuchenko et al. (2014); Gorshelev et al. (2014)) was less suited for ozone retrievals using the OMI-spectrometer than the BDM data, despite a similar spectral resolution ($0.01\,\mathrm{nm} - 0.018\,\mathrm{nm}$ for the $210 - 350\,\mathrm{nm}$ range) and a much better temperature coverage (data between $193\,\mathrm{K}$ and $293\,\mathrm{K}$ on a grid of $10\,\mathrm{K}$; see Weber et al. (2016) for example). Surprisingly, the same data set (SER) is now used to 'calibrate' the new BW data (see lines 95-99 of the manuscript):

   *Offset corrections were made for each of the 6 temperatures by fitting to the SER dataset since it was measured at higher ozone column density and thus considered more reliable regarding offset ... The offset corrections have minor effect on the cross-sections except for wavelengths above $\sim$330 nm.*

The procedure of dismissing the SER data set for ozone retreival, but using it for calibration is confusing and needs further explanation. The calibration procedure is even more surprising as the correction actually does not seem to impact the results of the present paper, because corrections are claimed to have minor effects within the OMI windows ($\leq 330\,\mathrm{nm}$). The necessity of making an offset correction arises from the measurement technique/setup at DLR. It thus needs to be explained why there is the need to make an offset correction in the first place and why the SER data do not suffer from the same problem.

2. In the introduction, the authors give the impression that new cross sections should be measured at a resolution of 0.01 nm or better. This contradicts the use of new cross section data that have been obtained at about 3 ($\lambda > 285.7$ nm) to 5 ($\lambda < 285.7$ nm) times lower resolution (see description of BW data set in section 2).

3. The authors use the terms Hartley and Huggins bands as well as OMI instrument windows to discuss different spectral regions in the UV. While wavelength ranges for both of the OMI UV windows are specified in the manuscript, no numbers are given for the Hartley and Huggins bands. Please indicate as this would help readers to follow the discussion.

4. There seem to be problems with the definitions of signs in some of the plots. For example, are the signs in Figure 7 correct? I find that local negative spikes in the total ozone column difference (BDM-BW) also correlate with cases where the tropospheric profile shows a tendency towards warmer colors (BDM > BW), which would indicate that either of the two scales (total ozone (TOC) vs altitude dependent ozone) should have a different sign. Another issue is the antarctic $+1\%$ BDM-BW bias in the TOC. From Figure 4, one would estimate that the cross section bias is positive when integrated all over the $(270 - 346)$ nm wavelength range (despite some few local negative spikes at low temperatures). This should result in a negative BDM-BW bias of TOC. Anyway, the antarctic positive TOC bias needs to be discussed as compared to the lower latitude value around $-1\%$ on the basis of the cross section data. In similar veins, the definition of the y-axis of Figure 4 shows that the room temperature BW cross-section is **negatively** biased with respect to BDM at low wavelengths. This is opposite to what is stated in line 254 of the manuscript *(Relative to the BDM data set, the BW data show systematic biases of $2-3\%$ in $C_0$ at shorter wavelengths below 300 nm, . . . ).*

5. In the comparison between BDM and BW in section 3, the BW data set is taken

as the baseline scenario. Because section 3 only provides a relative comparison and not an accuracy assessment, the authors should avoid the impression that BW is the truth (even though it compares more favorably with ozonesonde data presented in the next section 4). Instead of saying that BDM causes an underestimation or overstimation, it should just be stated that BDM estimates are lower or higher than estimates from BW.

6. Fig. 9 shows the OMI mean biases with respect to a common reference (ozonesonde). It would be nice to plot the reference profiles (or mean profiles with their sdev) along with the bias percentages.

7. TEMPO is not the only mission that will critically depend on refined ozone spectral data. IASI NG and UVNS are another example of combining retrievals in different domains. In the discussion, the authors need to mention/cite other ongoing or future activities on the synergistic use of different spectral regions that rely on the 9.6 $\mu$m region and the Chappuis band, eg. Costantino et al. (2017) and/or others.

**2 Technical**

**l. 32** *th → the*

**l. 95** indicate whether offset was assumed to be constant or wavelength dependent (for wavelength dependent offset specify dependence and range)

**l. 97** *(<270.27 nm) > and → (<270.27 nm) and*

**l. 106** *temperatures → temperature*

**l. 107** Should use terms $(T - 273.15\,\text{K})$ and $(T - 273.15\,\text{K})^2$ including the unit of K in eq. (1).

**l. 170**  *0.015 in UV1 → 0.015 nm in UV1*

**l. 254**  *BW data show systematic biases of 2-3 % in $C_0$ → BW data show systematic biases of 2-3 % in the cross section at $O°C$ ($C_0$)*

**l. 255**  *The difference in $C_1$ and $C_2$ implies distcinctly different → The differences in $C_1$ and $C_2$ imply a distinctly different*

**l. 268**  *200K → 200 K*

**l. 355**  list all author names

**l. 364**  *J. Quant. Spectrosc. Ra. → J. Quant. Spectrosc. Radiat. Transfer*

**p. 15**  Panels (a) - (c) should use logarithmic scales for the coefficients as BDM and BW curves are indistinguishable from 0 at wavelengths $\gtrsim$ 325 nm.

**p. 16**  Legend to Figure 3 should contain hint on the factor of five different scales used in panels (a) and (b)

**p. 17**  Legend to Figure 5 should better describe what is on the plot.

**p. 19 & 22**  Degree symbol $°$ before $K$ in x-axis legend of Figure 9 needs to be deleted. The same holds for the lower colour legend in Figure 7.

**p. 21**  Annotations *MB* and *MB $\pm$ SD* in upper right panel are misleading (there is no mean bias in the temperature plot). The 294 K temperature line for the BDM temperature point is drawn differently (thicker, other colour) than the other temperature lines.

**References**

Costantino, L., Cuesta, J., Emili, E., Coman, A., Foret, G., Dufour, G., Eremenko, M., Chailleux, Y., Beekmann, M., and Flaud, J.-M.: Potential of multispectral synergism for observing ozone pollution by combining IASI-NG and UVNS measurements from the EPS-SG satellite, Atm. Meas. Tech., 10, 1281–1298, 2017.

Gorshelev, V., Serdyuchenko, A., Weber, M., Chehade, W., and Burrows, J. P.: High spectral resolution ozone absorption cross-sections – Part 1: Measurements, data analysis and comparison with previous measurements around 293 K, Atmos. Meas. Tech., 7, 609–624, 2014.

Liu, C., Liu, X., and Chance, K.: The impact of using different ozone cross sections on ozone profile retrievals from OMI UV measurements, J. Quant. Spectroscop. Radiat. Transfer, 130, 365 – 372, HITRAN2012 special issue, 2013.

Serdyuchenko, A., Gorshelev, V., Weber, M., Chehade, W., , and Burrows, J. P.: High spectral resolution ozone absorption cross- sections – Part 2: Temperature dependence, Atmos. Meas. Tech., 7, 625–636, 2014.

Weber, M., Gorshelev, V., and Serdyuchenko, A.: Uncertainty budgets of major ozone absorption cross sections used in UV remote sensing applications, Atmos. Meas. Tech., 9, 4459–4470, 2016.

---

## Author Comment (AC1) · 10 Jul 2020

**General comments:**

This manuscript, AMT-2020-94, reports the evaluation of new O3 xsec data sets (labeled as "BW") measured in the Hartley and Huggins bands for the use of O3 profile retrieval from OMI observations. The BW data sets were modeled by using a polynomial in a function of temperature in order to facilitate direct comparison with the current reference data set ("BDM") and their application to the O3 profile retrieval. They have found that the new data set, BW, shows a better performance in the retrieval of O3 profile in terms of less oscillatory features in the retrieved profile and better agreement with the ozonesonde data. We found the manuscript written in a nice and compact manner; the presentation looks consistent. However, we are not convinced that we can agree with the authors' the interpretation of what is described in Sec. 2, which will be detailed below.

This manuscript has shown well that the new dataset, BW, is better than the BDM in the O3 profile retrievals primarily because of their wider temperature coverage, esp. going down to 194 K critical to the retrievals in the transition layers (UTLS), which was not covered by the BDM dataset in temperature. Therefore, the conclusion of this work has been supported by the results presented in the manuscript. The topics of this paper highly relevant to the scope of AMT, so that we recommend a publication of this manuscript to AMT with a revision or a further clarification Sec. 2. Specific comments and suggestions follow.

**Responses to general comments**

We would like to thank this reviewer for the constructive comments. We did our best to sincerely reply to 4 comments made by this reviewer.

**Specific comments**

**C1**. The authors wrote "Offset corrections were made for each of the 6 temperatures by fitting to the SER dataset since it was measured at higher ozone column density and thus considered more reliable regarding offset". Does this mean that the BW xsec was normalized to that of SER. Clarify what the corrections factors were and how (and what wavelengths) they were determined. Was this offset considered in the error budgets?

**R1**. Offset errors in the baseline of the measured spectra cause offset errors in the absorption cross section. Since the column amount of the ozone was limited by the relatively small absorption path of 22.1, the offset error in the ACS was relatively large, up to 2e-22 cm^2/molec. Around 344 nm this amounts to about 20% of the ACS. At 330 nm the offset is about 4%. At 270 nm the offset is about 0.0025%. In order to correct this error fits of the BW ACS to the SER ACS fitting a scalar and an offset were performed in the range 317-350 nm. The offset error in the SER ACS were much smaller due to the significantly longer absorption path (270 cm). The scalar was ignored. The offset was used to correct the entire wavelength range, but it would not have made a difference if we had limited it to the fit range since the offset error influence below 330 nm is negligible. The offset uncertainty was determined from the standard deviation of the fit multiplied with chi since the residuals were not purely noise. The offset uncertainty was 1e-24 cm^2/molec, which is negligible. We think that this discussion is beyond the scope of this paper, which is not intended, for developing/introducing this spectroscopic data, but for applying this dataset on our retrievals. The related discussion will be addressed in a separate paper lead by the author of this dataset, Manfred Birk.

**C2**. Author wrote, "After offset correction polynominals of $1^{st}$ order (<270.27 nm) and $2^{nd}$ order (>270.27 nm) in temperature were fitted for each spectral point to improve the statistical uncertainty" and followed by "Measured cross-sections are typically parameterized quadratically to be applied conveniently at any atmospheric temperatures" using the following equation: $C = C_o + C_1(T - 273.15) - C_2(T - 273.15)^2$.

**C2-1.** The agreement between the original data and the fitted data should be inspected or discussed for each of the two data sets, BW and BDM, and discussed. Besides, direct comparison of their original

data sets between BW and BDM (prior to having them fitted to the polynomial), which may be done at T = 273 and 295 K provided that their temperature differences, $\Delta = 0.5$ and 0.7K, respectively, is insignificant, which seems true because the authors argued the dominant coefficient C0 is almost independent of temperature.

**R2-1**. As mentioned in Section 2, the temperature correction has already been applied in the BW dataset available to the public. This paper is devoted to atmospheric validation of the BW dataset, rather than presenting the dataset itself. We think that it is out of scope to give a detailed evaluation for the original BW dataset where either offset and temperature correction is turn off because it is not officially published. The detailed views on the original/corrected BW dataset will be provided in another paper written by Birk and Wagner. In the ozone profile algorithm the cross sections parameterized using this quadratic equation are typically used to represent the dependence of cross-section on the atmospheric temperature vertically rather than the interpolated spectrum from original measurements. Therefore, this paper focused on comparing coefficients and the parameterized cross-sections between BDM and BW datasets.

**C2-2**. We are not sure how well the Eq. (1) could have captured the temperature dependence of the xsec. The xsec can be represented by integrated (line) intensities for the given frequency (wavelength) grid, and the temperature dependence of the line intensities can be modeled by two parameters, i.e., partition function (which we know well for O3) and the lower state energies (which we do not know for the features of this work). Thus, one can simulate the intensity ratio to that at 296 K at various temperature for a few representative cases of the lower state energies, as shown in Fig. X below. As we see, Fig. X is similar to the right panel of Fig. 1, except for one thing that each curve in Fig. X represents different values of the lower state energies, not the wavelength presented in Fig. 1. There is a possibility of having the sampled wavelengths (such as 280, 290,..., in nm) possessing progressively higher value of their (effective) lower state energies more appropriate to assume that each curve in Fig. 1 corresponds to a different of multiple transitions falling into the particular wavelength data point grid (for instance, 280nm±resolution element). This point should be addressed properly to keep naive readers from being misled to think the temperature dependences in Fig. 1 is attributed to the wavelengths.

**R2-2**. The quadratic equation was first found to represent well the temperature dependence of ozone cross sections in the UV [Paur and Bass, 1985] and has now become the standard approach [Liu et al., 2007;2013; Chehade et al., 2013a,b; Serdyuchenko et al., 2014]. In addition, Fig. X (this reviewer plotted) and Fig. 1 in this paper commonly imply that the dependence of the cross-section on the temperature tends to be linear at shorter wavelengths and slightly non-linear at longer UV wavelengths. Therefore, the quadratic (2nd) polynomials seem to be adequately fit the cross-section measurements. In revised manuscript, this discussion has been better specified by adding "This quadratic equation was first found to represent well the temperature dependence of ozone cross section in the UV (Paur and Bass, 1985) and has now become the standard approach (Liu et al., 2007; 2013; Chehade et al., 2013a;2013b; Serdyuchenko et al., 2014)" after the equation 1. The approach suggested by the reviewers is somewhat similar to pseudolines that is sometimes employed in the parametrizing the IR cross-sections, where temperature and pressure-dependent cross-sections are fit to a HITRAN-like line list where "transitions" do not have quantum mechanical meaning but do reproduce cross-sections. However, this approach is a lot more sophisticated than suggested by the reviewers because there are more than one transitions (with different intensities and lower state energies) that underlie absorption at selected wavelength. This very non-trivial and intense task has never been applied to the electronic spectra yet.

[Figure]

**Fig. X** S/S(296K) vs. T at a given E''(cm$^{-1}$) =[100,400,700,1000, 1300, etc.]

**C2-3** For the same reason, Fig. 2 is hard to interpret. The respective comparison of the C1 and C2 for two different data set as a function of nm could be legitimate only when the two data sets are measured at the same resolution because the effective lower state energies mentioned above would be the same. Therefore, the non-wavy feature of C2 for the BW data set would have more to do with the outcome of the resolution choice in the representation by Eq.(1), rather than it is telling the BW data set is superior to the BDM dataset in the temperature consistency. In other words, Fig. 2 shows which data set is better represented by Eq. 1 rather than which data is closer to the truth. This section may stay, but with a specific statement, being provided for the readers on the point made above. The bottom line is that the BW data set is better than the BDM set because of the broader coverage of the measurement temperature, especially covering the temperature critical to UTLS layers, as was properly concluded by the authors in the manuscript.

**R2-3.** We agree with this comment; it could be not straightforward to compare the coefficients especially C1 and C2 derived from BDM and BW, respectively, due to different spectral resolutions and the strong correlation between C1 and C2 especially when the temperature dependence is weak. However, important insights are obtained from this figure; the comparison of $C_o$ indicates systematic biases between two datasets, by 2 % on average, with some spikes of up to 8 % at longer UV wavelengths above 315 nm mainly due to the different spectral resolution. The $C_1/C_2$ characterizes the linear/non-linear dependence of the cross-sections. As shown in Figure 3.c, the quadratic temperature dependence show different behaviors in 290-310 nm, which is significantly correlated with the comparison of cross-section spectrum shown in Figure 4.

[Figure]

Revised Figure 4.

---

## Author Comment (AC2) · 10 Jul 2020

**General comments:**

The manuscript AMT-2020-94 provides a comparison of UV ozone retrievals from the OMI instrument using a new cross section data set (BW, provided in the frame of the ESA SEOM-IAS project) with the standard data set from Reims (BDM). Overall, the manuscript is very well written, nicely structured and argued. Selected figures do well illustrate the discussion in the manuscript. The presentation is scientifically sound and clear. The topic fits nicely within the journal scope and, therefore, I can fully recommend publishing the manuscript. There are a few issues to the current paper that need to be addressed before publication, however.

**Responses to general comments**

 We would like to thank this reviewer for the constructive comments. All the comments made by this reviewer were addressed in the revised manuscript.

**C1**. The analysis is based on a new cross section data set (BW data) that at this point of time is openly available, but has not yet been published in the scientific literature. It therefore lacks yet the scrutiny of the peer-review process. While this is a regrettable fact, it does not invalidate the present work. But the authors must carefully discuss what might possibly be an inherent contradiction. In a previous study (Liu et al., 2013), the authors have concluded that another recent UV cross-section data set (the SER data from Bremen, Serdyuchenko et al. (2014); Gorshelev et al. (2014)) was less suited for ozone retrievals using the OMI-spectrometer than the BDM data, despite a similar spectral resolution (0.01nm − 0.018nm for the 210 − 350nm range) and a much better temperature coverage (data between 193 K and 293 K on a grid of 10 K; see Weber et al. (2016) for example). Surprisingly, the same data set (SER) is now used to 'calibrate' the new BW data (see lines 95-99 of the manuscript): Offset corrections were made for each of the 6 temperatures by fitting to the SER dataset since it was measured at higher ozone column density and thus considered more reliable regarding offset. The offset corrections have minor effect on the cross-sections except for wavelengths above 330 nm. The procedure of dismissing the SER data set for ozone retrieval, but using it for calibration is confusing and needs further explanation. The calibration procedure is even more surprising as the correction actually does not seem to impact the results of the present paper, because corrections are claimed to have minor effects within the OMI windows (>330 nm). The necessity of making an offset correction arises from the measurement technique/setup at DLR. It thus needs to be explained why there is the need to make an offset correction in the first place and why the SER data do not suffer from the same problem.

**R1**. Offset errors in the baseline of the measured spectra cause offset errors in the absorption cross section. Since the column amount of the ozone was limited by the relatively small absorption path of 22.1 cm the offset error in the ACS was relatively large, up to 2e-22 cm^2/molec. Around 344 nm this amounts to about 20% of the ACS. At 330 nm the offset is about 4%. At 270 nm the offset is about 0.0025%. In order to correct this error fits of the BW ACS to the SER ACS fitting a scalar and an offset were performed in the range 317-350 nm. The offset error in the SER ACS were much smaller due to the significantly longer absorption path (270 cm). The scalar was ignored. The offset was used to correct the entire wavelength range, but it would not have made a difference if we had limited it to the fit range since the offset error influence below 330 nm is negligible. The SER data used for the offset fit were at longer wavelength and measured with an FTS, too. The structure of the spectra in this region agreed well beside a scalar up to 1.03, depending on temperature. In the lower wavelength range the SER data were obtained using a grating spectrometer and there were distinct differences in the structure. The offset correction is only relevant when using ACS at longer wavelength (e.g. Brewer, Dobson). In the current paper, however, opaque regions at lower wavelength are of interest, where the impact of the offset is rather small. As addressed to the answer to comment 1 from the first review, this discussion is out of scope to be detailed in this paper.

**C2**. In the introduction, the authors give the impression that new cross sections should be measured at a resolution of 0.01nm or better. This contradicts the use of new cross section data that have been obtained at about 3 (> 285.7 nm) to 5 (< 285.7 nm) times lower resolution (see description of BW data set in section 2).

**R2**. The spectral resolution requirement is from Orphal et al. (2016): ozone cross-sections should be measured at high spectral resolutions (typically 0.01 nm in the ultraviolet-visible). So the citation of "a resoultion of 0.01 nm or better" is not accurate and is probably confused with the wavelength calibration requirement "the spectral wavelength) calibration must be very accurate, too (typically at least 0.01 nm)." So we change the text from "at least 0.01 nm" to "typically 0.01 nm." For the BW dataset, measurements are performed at a coarser resolution to cover the broad spectral range as a tradeoff or spectrally degraded in the post-processing to increase signal to noise ratio. Indeed, the spectral resolution of 3.3 cm$^{-1}$ may have caused a very small deterioration of the highly resolved spectral features occurring above 325 nm. The high resolution structures have only a very small contrast regarding the underlying broad features. The impact is expected to be small, especially in view of the low resolution of the remote sensing instruments.

**C3**. The authors use the terms Hartley and Huggins bands as well as OMI instrument windows to discuss different spectral regions in the UV. While wavelength ranges for both of the OMI UV windows are specified in the manuscript, no numbers are given for the Hartley and Huggins bands. Please indicate as this would help readers to follow the discussion.

**R3**. We has specified the bands in the revised manuscript where these bands are first mentioned such as "$C_o$ values are similar to each other in the Hartley band (< 310 nm) with relative biases of 2-3%. However, the Huggins band (> 310 nm) shows large spiky biases of up to 8%. $C_1$ and $C_2$ represent linear and quadratic temperature dependences of absorption cross-sections, respectively"

**C4**. There seem to be problems with the definitions of signs in some of the plots. For example, are the signs in Figure 7 correct? I find that local negative spikes in the total ozone column difference (BDM-BW) also correlate with cases where the tropospheric profile shows a tendency towards warmer colors (BDM > BW), which would indicate that either of the two scales (total ozone (TOC) vs altitude dependent ozone) should have a different sign. Another issue is the Antarctic +1% BDM-BW bias in the TOC. From Figure 4, one would estimate that the cross section bias is positive when integrated all over the $(270-346)$ nm wavelength range (despite some few local negative spikes at low temperatures). This should result in a negative BDM-BW bias of TOC. Anyway, the antarctic positive TOC bias needs to be discussed as compared to the lower latitude value around $-1\%$ on the basis of the cross section data. In similar veins, the definition of the y-axis of Figure 4 shows that the room temperature BW cross-section is negatively biased with respect to BDM at low wavelengths. This is opposite to what is stated in line 254 of the manuscript (Relative to the BDM data set, the BW data show systematic biases of $2-3\%$ in C0 at shorter wavelengths below 300 nm, . . .).

**R4**. The contour map gives an impression that applying BDM causes the overestimation, especially around the tropopause where the coldest temperature/the lowest ozone amount is found. The impact of applying different cross-section dataset on total ozone retrievals are overwhelmed mainly by the lower stratospheric layers where the ozone amount is relatively large and the dependence of ozone-cross sections on the temperature is relatively important. Please take a look at the revised Figure 7 also including the contour map for absolute differences in the unit of DU (Figure 7.b), which shows that applying BDM causes the significant negative biases in the lower stratosphere (20-30 km) and then total ozone columns are underestimated. On the other hand, the BDM based total ozone columns are overestimated in South Pole due to the biggest inconsistency of two cross-sections at the coldest temperatures just above the tropopause. In the revised manuscript, this part has been better specified in page 6 as following:

  Figure 7 shows both relative and absolute differences of the retrieved ozone profiles with the corresponding temperature profiles taken from the National Centers for Environmental Protection (NCEP) final (FNL) operational global analysis data. Large differences of 20-50% commonly exist along the tropopause, where the original BDM measurements could not cover atmospheric temperatures below 218 K (Fig. 7a). Some larger differences occur throughout the troposphere in the tropics likely due to the relative smaller retrieved partial ozone columns. The individual differences of retrieved ozone in the lower troposphere are ~ 20%. However, the corresponding impact on the total column ozone, from integrating retrieved ozone profiles are overwhelmed by the stratospheric layers (20-30 km), as shown in Fig. 7b, where the ozone amount is relatively large and the dependence of ozone-cross sections on the temperature is still important. As a result, applying BDM causes an underestimation of total

ozone except at the South Pole due to the biggest inconsistency of two cross-sections at the coldest temperature just above the tropopause in spite of smaller amount of ozone compared to upper stratospheric layers. The magnitude of this underestimation/overestimation is ~1 %, which is comparable to the overall accuracy (~1.5%) of the OMI operational total ozone product against ground-based measurements (McPeters et al., 2015).

[Figure]

**Figure 7 in the revised manuscript.**

**C5**. In the comparison between BDM and BW in section 3, the BW data set is taken as the baseline scenario. Because section 3 only provides a relative comparison and not an accuracy assessment, the authors should avoid the impression that BW is the truth (even though it compares more favorably with ozonesonde data presented in the next section 4). Instead of saying that BDM causes an underestimation or overestimation, it should just be stated that BDM estimates are lower or higher than estimates from BW.

**R5**. We agree with this comment. The manuscript has been revised to reflect this suggestion.

**C6**. Fig. 9 shows the OMI mean biases with respect to a common reference (ozonesonde). It would be

nice to plot the reference profiles (or mean profiles with their sdev) along with the bias percentages.
**R6**. We have revised Figure 9, according to this comment. The revised figure is following:

[Figure]

**Figure 9.**

**C7**. TEMPO is not the only mission that will critically depend on refined ozone spectral data. IASI NG and UVNS are another example of combining retrievals in different domains. In the discussion, the authors need to mention/cite other ongoing or future activities on the synergistic use of different spectral regions that rely on the 9.6 μm region and the Chappuis band, eg. Costantino et al. (2017) and/or others.
**R7**. Yes, there are many on-going projects requiring the advanced ozone spectral data. However, the ozone profile algorithm used in this paper is optimized to retrieve ozone profiles from OMI BUV measurements with the capability of processing GOME, OMPS, and GOME/2 measurements, commonly focusing on the Hartley and Huggins bands. Furthermore, the TEMPO ozone profile algorithm has been under development by extending this OMI algorithm from UV only to UV+Visible. There have been several studies including this paper to recommend the reference ozone spectral data for UV spectral fitting, but nothing for the Chappuis band. Therefore, in the last section of this paper we addressed the importance about evaluating the visible ozone cross-section datasets, focusing on the SER and BDM datasets, which is one of priorities in the development of the TEMPO ozone profile algorithm. In this context, we think that it is out of scope to address other missions employing the thermal IR.

**2. Technical**

**C1**. (L32) th → the
**R1**. It has been revised.

**C2**. (L95) indicate whether offset was assumed to be constant or wavelength dependent
(for wavelength dependent offset specify dependence and range)
**R2**. The associated sentence has been revised for clarification from "Offset corrections were made for each of the 6 temperatures by fitting to the SER dataset" to "Offset corrections were made for each of the 6 temperatures by fitting to the SER dataset (constant for all wavelengths)"

**C2**. (L97) (<270.27 nm) > and → (<270.27 nm) and
**C3**. (L106) temperatures → temperature
**C4**. (L107) Should use terms $(T - 273.15K)$ and $(T - 273.15K)2$ including the unit of K in eq. (1).
**C5**. (L170) 0.015 in UV1 → 0.015nm in UV1
**C6**. (L254) BW data show systematic biases of 2-3% in $C_0$ → BW data show systematic biases of 2-3% in the cross section at O°C ($C_0$)
**C7**. (L255) The difference in C1 and C2 implies distcinctly different → The differences in C1 and C2 imply a distinctly different
**C8**. (L268) 200K → 200 K
**C9**. (L355) list all author names
**C10**. (L364) J. Quant. Spectrosc. Ra. → J. Quant. Spectrosc. Radiat. Transfer
**R2-R10**. We accepted all these suggestions.

**C11** (p. 15) Panels (a) - (c) should use logarithmic scales for the coefficients as BDM and BW curves are indistinguishable from 0 at wavelengths $\geq$ 325 nm.
**R12**. We revised Figure 2 to use logarithmic scales in y-axis.

**C12** (p. 16) Legend to Figure 3 should contain hint on the factor of five different scales used in panels (a) and (b).
**R12**. In caption, it was detailed like "In the legend, the temperatures not covered by each dataset are indicated with gray and black, for values beyond lower and upper boundaries, respectively", but we added "$T > T_{max}^{BDM}$ $T < T_{min}^{BDM}$" in Fig. 3 a and "$T > T_{max}^{BW}$ $T < T_{min}^{BW}$" in Fig. 3. b according to this comment.

**C13** (p. 17) Legend to Figure 5 should better describe what is on the plot.
**R13**. For clarification, the caption has been revised like "The impact of parameterizing the cross-sections shown in Figure 3 on ozone profile retrievals, for (a) BDM and (b) BW, as a function of solar zenith angle (SZA). The differences of retrieved ozone profiles are assessed in absolute (left panels) and relative (right panels) units, respectively."

**C14** (p. 19) & 22 Degree symbol ° before K in x-axis legend of Figure 9 needs to be deleted. The same holds for the lower colour legend in Figure 7.
**R14**. °K has been corrected to K in indicated figures.

**C15** (p. 21) Annotations MB and MB ± SD in upper right panel are misleading (there is no mean bias in the temperature plot). The 294 K temperature line for the BDM temperature point is drawn differently (thicker, other colour) than the other temperature lines.
**R15**. This figure has been replotted after correcting indicated annotations and line.